# Microglial Endocannabinoid Signalling in AD

**DOI:** 10.3390/cells11071237

**Published:** 2022-04-06

**Authors:** Lucia Scipioni, Francesca Ciaramellano, Veronica Carnicelli, Alessandro Leuti, Anna Rita Lizzi, Noemi De Dominicis, Sergio Oddi, Mauro Maccarrone

**Affiliations:** 1Department of Biotechnological and Applied Clinical Sciences, University of L’Aquila, Via Vetoio Snc, 67100 L’Aquila, Italy; lucia.scipioni@graduate.univaq.it (L.S.); veronica.carnicelli@univaq.it (V.C.); annarita.lizzi@univaq.it (A.R.L.); noe.dedominicis@gmail.com (N.D.D.); 2European Center for Brain Research/IRCCS Santa Lucia Foundation, Via del Fosso di Fiorano 64, 00143 Rome, Italy; fciaramellano@unite.it (F.C.); a.leuti@unicampus.it (A.L.); 3Faculty of Veterinary Medicine, University of Teramo, Via R. Balzarini 1, 64100 Teramo, Italy; 4Department of Medicine, Campus Bio-Medico University of Rome, Via Alvaro del Portillo 21, 00128 Rome, Italy

**Keywords:** endocannabinoid, microglia, Alzheimer’s disease, neuroinflammation

## Abstract

Chronic inflammation in Alzheimer’s disease (AD) has been recently identified as a major contributor to disease pathogenesis. Once activated, microglial cells, which are brain-resident immune cells, exert several key actions, including phagocytosis, chemotaxis, and the release of pro- or anti-inflammatory mediators, which could have opposite effects on brain homeostasis, depending on the stage of disease and the particular phenotype of microglial cells. The endocannabinoids (eCBs) are pleiotropic bioactive lipids increasingly recognized for their essential roles in regulating microglial activity both under normal and AD-driven pathological conditions. Here, we review the current literature regarding the involvement of this signalling system in modulating microglial phenotypes and activity in the context of homeostasis and AD-related neurodegeneration.

## 1. Introduction

Compelling genetic and pathological evidence strongly supports the “amyloid cascade hypothesis” of AD, which states that beta-amyloid (Aβ), and in particular the least soluble 42-amino-acid-long Aβ isoform, is the causative agent in the pathogenesis of all forms of AD [1,2]. However, Aβ appears to be a necessary, yet insufficient, factor for AD development [3]. Indeed, not all elderly patients who have significant amyloid plaque pathology develop the disease or show cognitive impairment, indicating that other ageing-promoting processes, by compromising the brain’s capacity to respond to Aβ peptides adequately, can determine the pathogenic evolution of amyloidosis [3]. Recent evidence supports the idea that microglial dysregulation is indeed a crucial driver in the pathogenesis of AD [4]. Altered/dystrophic microglial cells could contribute to aggravating and propagating an Aβ-pathologic cascade throughout the brain [5,6] by acquiring a “pro-AD” phenotype that consists of: (*i*) the chronic release of pro-inflammatory cytokines and other inflammatory mediators, such as reactive oxygen and nitrogen species; (*ii*) reduced phagocytosis and Aβ clearance; (*iii*) reduced release of neurotrophic factors; (*iv*) reduced release of pro-resolving factors; and (*v*) reduced motility (chemotaxis).

The immune system can be modulated and regulated by humoral factors and metabolic products. Among these, eCBs are bioactive lipids that increase or decrease distinct immune functions when mobilized at the very beginning or shortly after first-line immune modulators [7]. Notably, microglia express the array of receptors and metabolic enzymes (collectively termed “ECS”) that control the immune-related functions of eCBs [8]. This system displays a wide distribution throughout the body and is involved in many adaptive responses to stressful internal and/or environmental insults. During the last few years, the brain ECS—by virtue of its capability of orchestrating neuromodulatory, anti-excitotoxic, anti-inflammatory/pro-resolutive, and anti-oxidative actions—has emerged as a key player in several neurodegenerative disorders, including AD [9].

In this review, we highlight the relevance of eCBs signalling in microglial physiology and its therapeutic potential to maintain/restore microglial homeostatic functions in the context of AD. A brief overview of the ECS is presented to summarize its key components and the different signalling pathways that it can elicit. Then, the possible roles of eCBs in microglial biology are discussed, describing how these bioactive lipids may regulate the cellular processes involved in the homeostasis of the normal and diseased brain, such as phagocytosis, chemotaxis, cytokine/neurotrophic factor release, microglial differentiation, and activation. Finally, current evidence that supports the idea that targeting microglial ECS may represent a valuable disease-modifying strategy for AD is briefly described and commented on.

## 2. Microglia

Microglia, the resident immune defence in the brain, modulates the development, activity and plasticity of the central nervous system (CNS). To perform these complex functions, microglia adopt different activation states/phenotypes, depending on the microenvironment, which engage them in neuroinflammation, tissue repair, and even pro-resolutive inflammatory processes [10,11].

### 2.1. Microglial Functions and Phenotypes

Microglia represent highly versatile cells that play a pivotal role in the neurobiological, neuroinflammatory and neurophysiological homeostasis of the CNS in both health and disease. Often referred to as brain-resident macrophages—a definition that has been often criticized due to the ontogenetic and functional differences between these two cell types—microglia maintain neuroinflammatory homeostasis by interacting with other immune cells (e.g., T cells) and by releasing a vast array of pro- and anti-inflammatory cytokines and endogenous lipids, such as eCBs, arachidonic acid-derived autacoids and pro-resolving mediators [12,13,14]; on the other hand, they also participate in pivotal brain functions, such as the elimination of dead neurons and cell debris [14], synapse pruning and, to a minor extent, the regulation of the synaptic neurotransmitter tone.

In general, “resting” microglial cells exhibit a highly ramified appearance that gives them the ability to constantly survey the surrounding brain parenchyma, as they look for the presence of danger-, pathogen- or resolution-associated molecular patterns (DAMP, PAMP and RAMP, respectively) through their vast array of recognition systems represented by Toll-like receptors (TLRs), Nod-like receptors (NLRs), C-type lectin receptors (CLRs) and RIG-like receptors (RLRs), altogether referred to as pattern recognition receptors (PRRs) [15,16]. In particular, PRRs expressed on microglia surfaces recognize any molecular signal that indicates damage or cell stress, including ATP, nucleic acids, necrotic cells and cell debris and, relevant to this review, misfolded proteins such as Aβ species [14,15]. The recognition of any of these ligands leads to their internalization and the activation of microglia. Although in different neuroinflammatory and neurodegenerative paradigms, microglial cells have been reported to sport quite a vast array of morphologies, their pathological activation generally results in the transition towards an amoeboid phenotype that triggers potent neuroinflammatory reactions through the secretion of cytokines and chemokines [17,18]. Furthermore, depending on the nature of the recognized stimulus and/or the surrounding milieu to which they are exposed, microglia can assume different profiles, or phenotypes, the equilibrium of which contributes to the onset and outcome of neuroinflammatory and neurodegenerative mechanisms. 

#### 2.1.1. Microglial Phenotypical Heterogeneity

The past three decades have seen a quite lively debate about the different states that microglial cells can display under physiological and pathological conditions. A common classification that has been widely applied to describe microglia phenotypes—one that mimics the approach used to describe pro- and anti-inflammatory macrophages—involves the use of the M1–M2 dichotomy to describe detrimental or neuroprotective states of microglial cells, respectively [19]. On the other hand, although a number of microglial markers have been associated with neuroinflammatory or pro-homeostatic properties of these cells, pieces of evidence suggest that microglia exist in vivo as a rather blurred distribution of states, two of which are described by the M1–M2 paradigm [20,21].

Relevant to this, although a few surface markers are commonly utilized in several panels to characterize microglia—first and foremost, CD11b, CD68 and HLA-DR—these lack the specificity to differentiate between the discrete states that the M1–M2 dichotomy is supposed to describe (mostly because they can be expressed both on M1- and M2-like cells, though at different levels) [22]. On the other hand, other markers are currently used to assess the immunophenotypical distribution of microglial cells, including CD14, CD11c, CD16/32, CD33, CD36, CD64, CD163, CD206, CD200 or the triggering receptor expressed on myeloid cells 2 (TREM2) [17,22,23]. However, their use to univocally distinguish between different microglial profiles, especially in light of the large immunophenotypical and morphological heterogeneity that these cells exhibit, even in different brain locations, is still debated. Unfortunately, the presence of significant differences between human- and mouse-derived cells—a common example being represented by the exclusive presence of F4/80 in mouse microglia and the still-debated expression of arginase-1 in human tissues [23]—adds a further layer of complexity to the picture. Finally, a number of genome-wide expression profiling studies have failed to individuate sufficient diversity in the microglia population to reduce their diversity to two phenotypes [21,24]; nonetheless, M1- and M2-like markers are still widely used as a tool to describe the neuroimmune properties of these cells in several neuroinflammatory contexts, including AD. Although in vitro these phenotypes are largely represented in this pure form, classifying these cells in vivo has proven to be more challenging, reflecting the plastic nature of microglia [25].

##### Pro-Inflammatory-M1-like Microglia

Pro-inflammatory, often referred to as M1-like, microglia are induced by tissue injury and phlogistic soluble factors. In particular, the presence of bacterial components (e.g., LPS) and cellular debris, as well as of IL-1β, IL-8, IFNγ and TNF in the surrounding milieu, induces the activation inflammasome complexes, such as NOD-like receptor family pyrin 1 and 3 (NRLP1 and 3) and, consequently, the release of neuroinflammatory factors, such as the pro-inflammatory cytokines (e.g., IL-1α/β, IL-6, IL-23 and TNF), chemokines (e.g., CCL2, CXCL9, CXCL10 and CCL20), co-stimulatory proteins (e.g., CD40 and MHC-II) and the activation of enzymes involved in the induction of oxidative stress (e.g., NADPH oxidase and the inducible nitric oxide synthase (iNOS)), which result in excitotoxicity, neurotoxicity, demyelination and, ultimately, neuronal dysfunction and death [14,26,27]. Alongside the induction of the aforementioned factors, M1-like polarization is often accompanied by an upregulation of CD11b, CD16/32, CD68 and CD86, with the concomitant activation of typical transcription factors, also previously described for peripheral macrophages, such as NF-kB, STAT1 and STAT3 [28]. 

Interestingly, similar to monocyte-derived macrophages, polarization in microglia also induces dramatic changes in the metabolic profile [26,29]; indeed, whereas microglial cells seem to mostly rely on oxidative phosphorylation under a surveillance state [25,30], M1 polarization drives these cells towards glycolytic/anaerobic metabolism and the consequent production of lactate and reactive species (as reviewed in [26]).

##### Anti-Inflammatory-M2-like Microglia 

M2-like microglia possess pro-homeostatic properties and are thought to play a role in the resolution processes. In particular, these cells play a crucial activity in the elimination of cell and myelin debris, dead cells and misfolded proteins that, if not cleared from the brain parenchyma, are primary inducers of neuropathology and degeneration [31], as also suggested by the fact that highly phagocytic microglial cells can be found in damaged brain tissue [32]. On the other hand, defective or dysfunctional microglial phagocytosis has been observed in mouse models of neuropathological or neurodegenerative conditions, such as ischemic stroke, multiple sclerosis and AD [33,34,35,36]. M2-like polarization in microglia is similarly driven by peripheral M2 macrophages and anti-inflammatory/TH2-derived cytokines, such as IL-4 and IL-13 [26], with ω-3- and ω-6-derived specialized pro-resolving lipids (SPM) (i.e., resolvins, protectins, maresins, and lipoxins) that enhance their pro-homeostatic properties while blunting the production of pro-inflammatory mediators [30,37]. In particular, at a molecular level, M2-like polarizing stimuli act by inducing NF-kB-antagonizing transcription factors, such as STAT6, which, in turn, trigger the production of anti-inflammatory cytokines, such as IL-4, IL-10 and TGFβ, as well as chemokines such as CCD22 [14,28] and the expression of anti-oxidant factors, such as nuclear factor erythroid 2-related factor (Nrf2) [38]. Murine cells also display an upregulation of arginase 1 (Arg1), an enzyme that counteracts inflammatory and oxidative damage both through the arginine-dependent biosynthesis of tissue-regenerating polyamines, as well as through the reduction in NO levels, which is achieved by outcompeting iNOS for the availability of their common substrate, arginine [25]. On the other hand, M2-like microglia are characterized by high expressions of CD206 and CD36, two scavenger receptors involved in the phagocytosis of cell and myelin debris [28,39], and by the production of neurotrophic factors, such as the nerve-derived neurotrophic factor (NGF), glial cell-derived neurotrophic factor (GDNF) and brain-derived neurotrophic factor (BDNF) [40]; in mice, they are characterized by the expression of FIZZ1 and Ym1, two secretory proteins involved in tissue regeneration. Notably, a recent paper also reported a role of the STAT6/Arg1 axis in the induction of anti-inflammatory/pro-resolving functions such as efferocytosis (i.e., the process which, during the resolution of inflammation, leads to the removal of apoptotic cells and tissue debris) [41]. M1-like cells shift their metabolic state towards glycolysis while shutting down mitochondrial function, whereas M2 mostly rely on oxidative phosphorylation and respiration. This strongly suggests an integration of these several signals in determining the outcome of neuroinflammation in the CNS.

Several works in the past two decades have also described the existence of a number of M2 subtypes that can be obtained in vitro, in macrophages and microglia, through the stimulation with different immunomodulatory ligands: in this taxonomy, “canonical” IL-4/13-stimulated macrophages are defined M2a; on the other hand, M2b are induced by immune complex and TLR or IL1R ligands, while M2c is induced through IL-10/TGFβ [20,42]. In general, M2a cells express higher levels of CD206, TGFβ, Fizz1 and Arg1; M2b cells produce IL-10 as well as M-CSF and G-MCSF; M2c cells produce higher levels of scavenger receptors such as CD206, CD163, SR-A1 and B1. It should be noted that this classification describes “cell types” that are obtained in vitro under very controlled conditions, while it is quite likely that in vivo environments are characterized by less-defined phenomena, with microglial cells that display intermediate immunophenotypes.

### 2.2. Microglia and Alzheimer’s Disease

#### 2.2.1. General Traits of Alzheimer’s Disease

AD accounts for almost 2/3 of the worldwide cases of dementia and cognitive decline, with 1 out of 10 people over 65 years and almost 1 out of 3 over 85 years suffering from this condition worldwide [43]. It is characterized by massive neuronal death, cortical atrophy and progressive cognitive decline. Although a number of different environmental and genetic causes have been described in this pathology, the cause of the neurodegeneration has been historically linked to the aberrant accumulation of two hallmark protein aggregates, which represent pathognomonic signs of AD: (*i*) amyloid-β (Aβ) oligomers and polymers build up in the brain of affected people and constitute the typical plaques that are considered the main cause of neuronal loss and toxicity [44]; (*ii*) tau, a protein involved in stabilizing the microtubules and controlling axonal trafficking, undergoes pathological modifications that lead to its detachment from the microtubules and its aggregation to form tangles [45]. Even though amyloid plaques and tau tangles represent historical targets of investigation in this field, neuroinflammation has emerged in recent times as a central cause of neuronal loss in AD, with microglial cells considered to be a primary source in this process.

#### 2.2.2. The Involvement of Microglia in Alzheimer’s Disease

Microglia are profoundly involved in the correct functioning of the neuronal tissue, and overactivated or dysfunctional microglia are commonly featured in brain pathophysiology, cognitive decline, and AD, sometimes even preceding the neurological symptoms of the disease. In particular, microglia-induced neuroinflammation seems to represent a pivotal and necessary event in the development of the AD clinical phenotype. Indeed, microglial activation has been reported in patients with MCI- [45,46], even in the absence Aβ aggregates, and in animal models even before the formation of amyloid plaques [46,47,48]. Further rationale supporting the role of early neuroinflammation in AD is provided by several recent studies. For example, the inheritance of pathological alleles for genes that are strongly involved in innate and microglial responses (e.g., TREM2, CD33 and complement proteins) represent a genetic risk for the development of AD (as reviewed in [14]). Furthermore, microglia-mediated neuroinflammation has proven necessary, alongside aggregates, to induce the neuropathological features of the disease in animal models, and Aβ alone seems to be insufficient to induce symptoms [49]; however, autoptic samples of asymptomatic patients with high Aβ brain loads also did not display activated microglial cells [50]. The activation of microglia by Aβ contributed through the pyrin domain-containing 3 (NLRP3) inflammasome to enhance Aβ aggregation and to the tau progression [51,52].

Apparently, not only chronically-inflamed microglia is at the base of the clinical symptoms of AD, but the presence of Aβ seems to act as an enhancer, due to the fact microglial cells can recognize amyloid and tau aggregates as molecular profiles (i.e., PAMPs/DAMPs) through PRR-dependent signalling. This leads to the morphological changes of these cells and to the production of pro-inflammatory cytokines, such as IL-1β, IL-6, IL-8, TNF, as well as reactive oxygen species [14]. This transition towards disease-associated microglia (DAM), which occurs with the progression of the disease, is characterized by a distinct transcriptomic fingerprint, which features the downregulation of homeostatic genes (e.g., purinergic receptors, Cx3cr1 and Tmem119) and the concomitant upregulation of AD-related genes such as ApoE, cathepsin D (Ctps), lipoprotein lipase (LPL), tyro protein tyrosine kinase binding protein (TyroBP) and Trem 2 [40]. Of note, other receptors, such as formyl peptide receptor 2 (FPR2, also known as ALX), a G protein-coupled receptor that engages pro-resolving lipids, has been reported to bind Aβ [53]. FPR2 displays strong ligand-biased signalling, and is expressed in microglia [53], suggesting that AD pathology might also be related to interference between inflammation and resolution networks. Interestingly, the microglial pathology in AD also manifests in the form of aberrant phenotypes and morphologies, in addition to those that were previously described. Indeed, senescent brains display, in general, microglial cells with a lower level of branching [14], while other works have reported that the morphology of these cells can significantly change depending on their relationship with amyloid aggregates, with near-plaque microglia displaying altered morphology and electrophysiological properties as compared to other cells that have no contact with Aβ structures [54]. Of note, the study of microglia in cognitive decline has also led to the recent description and characterization of “dark microglia”, an AD-restricted phenotype described in mice prone to develop AD, which displays evident signs of oxidative stress and expresses high levels of CD11b, TREM2 and 4D4 [55]. These cells are strongly associated with the neurons of the hippocampus, cortex, amygdala and hypothalamus of affected mice, and completely encircle synapses at the level of both axonal terminals and dendritic spines, suggesting high levels of synaptic stripping that might be involved in the neuropathology of AD [55].

## 3. The Endocannabinoid System

The eCBs *N*-arachidonoylethanolamine (AEA) and 2-arachidonoylglycerol (2-AG) are lipid messengers acting as endogenous ligands for the cannabinoid receptors CB_1_ and CB_2_. Over the past two decades, these signalling lipids have emerged as critical mediators of many aspects of human health and disease, revealing a surprising organizational complexity in the mechanisms of synthesis, transport, and degradation. Although initially described as neuromodulators/neurotransmitters, these bioactive lipids also display multiple and relevant immunomodulatory properties [56,57].

### 3.1. eCBs Synthesis and Degradation

All cells in the body, including those of the immune system, produce AEA and 2-AG, the most studied eCBs. Rather than being pre-synthesized and stored in secretory vesicles, these bioactive lipids are made “on-demand” (i.e., when and where they are needed) by the receptor-stimulated cleavage of precursor membrane phosphoglycerides by several hydrolases.

AEA and 2-AG synthesis occurs through many alternative routes, which can also co-exist in the same cell and contribute to the production of eCBs in a time-, space- and activity-dependent manner [57,58]. AEA originates from a phospholipid precursor, *N*-arachidonoylphosphatidyl ethanolamine (*N*ArPE), which is, in turn, formed from the *N*-arachidonoylation of phosphatidylethanolamine via both Ca^2+^-sensitive and Ca^2+^-insensitive *N*-acyltransferases (NATs and iNATs) [59]. *N*ArPE is then converted into AEA by several possible alternative pathways, the most direct of which is catalyzed by an *N*-acylphosphatidylethanolamine-selective phosphodiesterase (NAPE-PLD) [60]. In macrophages and other immune cells, another alternative biosynthetic pathway for AEA involves the PLC-catalyzed cleavage of *N*ArPE to yield phospho-AEA, which is subsequently dephosphorylated by protein tyrosine phosphatase non-receptor type 22 (PTPN22), a member of PEST family of protein tyrosine phosphatases [61,62].

Through the hydrolysis of different arachidonoyl-containing membrane lipids, 2-AG synthesis can potentially occur. The best-studied synthetic route for 2-AG is its synthesis from *sn*-2-arachidonic-containing diacylglycerols (DAGs) [63] by one of two DAG lipases (DAGL) isoforms, DAGLα and DAGLβ [64]; the latter isoform is expressed more in macrophages, and although its relative brain expression is sparse, it is highly expressed in microglia [65]. Alternatively, 2-AG can also be synthesized by the dephosphorylation of *sn-2* arachidonoyl-lysophosphatidic acid (LPA) [66] via 2-LPA phosphatase; or by the sequential action of PLA1 and a lysophospholipase C (lyso-PLC) [67] in *sn-2* arachidonate-containing phosphatidylinositol (PI) and its derivative 2-arachidonoyl-lysoPI, respectively [68,69]. 

The biological effects of eCBs are terminated by cellular uptake followed by intracellular degradation. As very lipophilic compounds, AEA and 2-AG can passively diffuse through the membrane bilayer, even if this process seems to be accelerated by a rapid and selective carrier system (i.e., eCBs membrane transporter, EMT) that is postulated to be expressed in both neurons and glial cells [70]. Although there is strong indirect evidence for the existence of this transmembrane transport, the molecular identity of the protein(s) involved remains to be assessed. In this context, several lipid-carrier proteins, which assist the movement of eCBs within and outside cells, have been identified, confirming that, despite the current controversy, the eCBs transport system should be further characterized in the future [71,72].

Once internalized, eCBs can be hydrolyzed by distinct serine hydrolases. The major AEA catabolizing enzyme is the fatty acid amide hydrolase (FAAH), which releases arachidonic acid and ethanolamine [57,70]. 2-AG is mainly hydrolyzed by monoacyl-glycerol lipase (MAGL) [73] and, to some extent, by other enzymes, such as FAAH and the α/β-hydrolases (ABDH6) and 12 (ABDH12) [74]. Finally, AEA and 2-AG can be also metabolized via oxidation by lipoxygenase and cyclooxygenase (COX) enzymes [75].

### 3.2. eCBs Receptors and Molecular Pathways

CB_1_ and CB_2_ receptors are members of the superfamily A of the heptahelical transmembrane-spanning G protein-coupled receptors (GPCRs) coupled to heterotrimeric Gi/o proteins [76,77]. These receptors are expressed to various extents in immune cells, with CB_2_ being predominant under physiological conditions and upon acute and chronic inflammation [78]. Although CB_1_ is primarily expressed in specific neuronal populations, with neuromodulatory activity, it also seems to play a role in regulating immune responses and inflammatory pathways [79,80]. In this context, Ativie and colleagues demonstrated that neuronal CB_1_ may indirectly regulate microglial activity, possibly by influencing the crosstalk between neurons and microglia [81].

The binding of eCBs to CB receptors affects several cellular pathways, such as the inhibition of adenylate cyclase and then of protein kinase A (PKA); the regulation of ionic currents (inhibition of voltage-gated L, N and P/Q-type Ca^2+^ channels, activation of K^+^ channels); the activation of focal adhesion kinases, such as MAPKs (p38, ERK1/2, JNK), PI3K/Akt and cytosolic phospholipase A2; and the activation (CB_1_) or inhibition (CB_2_) of iNOS. All of these pathways are involved in fundamental microglia functions [57,82,83,84]. 

There is mounting evidence that eCBs also exert their biological activity via additional non-cannabinoid receptors, such as the transient receptor potential vanilloid type-1 (TRPV1) ion channel, which is functionally expressed in microglia [85,86,87,88]. Other non-canonical eCBs receptors expressed by microglia are the nuclear peroxisome proliferator-activated receptors (PPAR) α and γ, and the orphan G protein-coupled receptors GPR55 and GPR119 [56,89].

## 4. Microglial Endocannabinoid System in Alzheimer’s Disease

### 4.1. Role of the ECS in Microglial Functionality

The ECS has a pivotal role in brain inflammation by regulating microglial biology in terms of proliferation, migration, phagocytosis, and the production of pro- and anti-inflammatory mediators [8] (Figure 1).

#### 4.1.1. eCBs Receptors

Although CB_1_ was initially considered as a neuron-specific cannabinoid receptor, emerging evidence is revealing that this receptor is also constitutively expressed, even if at low levels, in microglia [90,91]. Indeed, a recent work with conditional knockout mice documented that hippocampal CB_1_^−/−^ microglia show a decreased expression of TNF-α compared to wild-type mice upon stimulation with LPS [92]. However, to date, except for some rare works, the specific role of CB_1_ in microglial cellular physiology has not been deeply explored [92,93,94].

CB_2_ immunoreactivity was primarily associated with astrocytes and microglial cells in the healthy brain. The expression of CB_2_ is significantly upregulated in these cells following brain trauma or under other pathological conditions, including AD [95], Parkinson’s disease [96], and multiple sclerosis [97]. Similar findings were observed in several mice models of neurodegenerative conditions [98,99]. The genetic ablation of CB_2_ showed microglia with a reduced phagocytic capacity and relevant morphology alterations during the switching in M2 phenotype compared to wild-type cells. In particular, M2a microglia from CB_2_^−/−^ mice lost well-defined and multiple lamellipodia and took a more elongated shape. In addition, Arg-1 expression was diminished in CB_2_^−/−^ both under basal conditions and following M2a stimulation, suggesting a role for CB_2_ in anti-inflammatory switching [90]. In rat microglia, CB_2_ activation by 2-AG leads to the stimulation of proliferation and endocytosis of the receptor. The effect was reversed by the antagonist of CB_2_ receptor, SR144528, and mimicked by the CB_2_ receptor-specific agonist JWH133 [100]; in this context, CB_2_ appeared to be the most relevant eCBs receptor acting on the release of pro- and anti-inflammatory mediators [101]. Indeed, the CB_2_ agonist AM1241 was shown to suppress the expression of pro-inflammatory cytokines, IL-1β, IL-6, iNOS, in LPS/INFγ-activated microglial cells [101]. At the same time, there was an increase in the expression of Arg1, IL-10, and the neurotrophic factors BDNF and GDNF, which were significantly reduced by the co-administration of the CB_2_ antagonist AM630 [101]. These findings are consistent with many other studies carried out on primary cells and several immortalized cell lines. When microglia are experimentally activated in a reactive state, CB_2_ activation could inhibit the release of pro-inflammatory and cytotoxic factors interfering with their switching to a neurotoxic phenotype [102,103,104,105,106].

In mouse brains, TRPV1 is primarily expressed in microglia [88], and its relevance in microglial physiology has only been highlighted in recent years. TRPV1^−/−^ mice challenged with LPS to induce systemic inflammation showed a better survival rate, accompanied by decreased microglial activation [87]. TRPV1 deletion in microglial cells inhibited NLRP3 inflammasome activation and capsaicin-induced migration [87]. Moreover, the induction of experimental autoimmune encephalomyelitis in TRPV1^−/−^ mice led to less inflammatory-cell infiltration, reduced Iba-1 expression and restored myelin damages [107]. Consistently, the stimulation of TRPV1 by capsaicin, in wild-type microglia, induced a pro-inflammatory phenotype with higher levels of TNF-α and lower levels of IL-10. The same stimulation in TRPV1^−/−^ microglia led to equal amounts of TNF-α, but significantly higher amounts of IL-10, as in the anti-inflammatory activation state of microglia [88]. Some discrepancy was found in vitro studies in which TRPV1 activation, rather than inactivation, reduced some cytotoxic factors, such as microglial-originated ROS [86,108].

#### 4.1.2. eCBs Metabolic Enzymes

Microglia express a full assortment of synthetic and catabolic enzymes for eCBs and can therefore metabolize both 2-AG and AEA [8,109]. Effectively, microglia produce in vitro 20 times more eCBs than neurons and other glial cells and are likely to be the primary cellular source of these bioactive lipids under neuroinflammatory conditions [110]. Moreover, eCBs metabolic enzymes seem to be regulated by microglial activation states [110,111]. In particular, stimulating the switch of microglia to either M2a or M2c states led to an upregulation of the biosynthetic enzymes (i.e., DAGLα in M2a; NAPE-PLD in M2c), accompanied by a downregulation of the respective degrading enzymes, thus resulting in the elevated production of eCBs [90].

The pharmacological inhibition of MAGL was shown to raise 2-AG level in the brain [112] and exerted beneficial immunomodulatory functions in neuroinflammatory conditions, attributable, at least in part, to the regulation of microglial functions. For example, in a PD mouse model, JZL184 (a selective MAGL inhibitor) prevented motor impairment and induced an increase in microglial cell number and ramification in the striatum, suggesting that MAGL activity impacted both microglial proliferation and phenotypes [113]. In a viral-induced neuroinflammation model, UCM03025 (a reversible MAGL inhibitor) and/or 2-AG administration act through CB_2_ to mediate long-term beneficial effects. In particular, in this neuroinflammatory model, MAGL inhibition hampered microglial cells in reaching an activated pro-inflammatory phenotype, as seen in a morphological analysis where microglia showed a reduction in complexity and reactivity comparable with their resting state [114]. MAGL inhibition (by JZL184) was established as preserving the neuronal function in experimental autoimmune encephalomyelitis and the non-immune demyelination model by reducing inflammation markers and suppressing microglial activation (assessed through CD11b immunoreactivity) [115]. Additionally, in a study of neuroinflammation induced by acute systemically administered LPS, MAGL pharmacological inhibition, as well as its genetic ablation, reduced cerebral pro-inflammatory cytokine release and microglial reactivity assessed through Iba-1 expression [116]. The latter study reported that the effect was not directly mediated through CB_1_ or CB_2_-dependent mechanisms. These findings suggested that the immunomodulatory effects on microglia observed under MAGL inhibition may have involved other non-cannabinoid receptors, such as PPARs [117].

The selective inhibition of FAAH exerted beneficial effects by modulating microglia responses under different types of pro-inflammatory stimuli. For example, chronic treatment with PF3845, a selective inhibitor of FAAH, in mice subject to traumatic brain injury suppressed the expressions of iNOS and COX-2, while it enhanced the expression of Arg-1 through phosphorylation of ERK1/2 and AKT. These findings suggest that AEA signalling may promote a shift of microglia from the M1 to M2 phenotype [118]. In a model of ethanol-induced neuroinflammation, URB597, another selective FAAH inhibitor, improved memory, reduced iNOS, TNF-α, IL-6, and monocyte chemoattractant protein-1 (MCP-1) and increased TLR4, concomitantly producing relevant morphologic changes in microglia [119]. Interestingly, the chronic administration of URB597 (and AEA) increased the number of Iba-1-positive cells in the hippocampus, both in ethanol and control groups [119]. These findings were in agreement with those of another study characterizing the immunophenotype of FAAH^−/−^ mice, where enhanced density and cell size of microglia were also found in the hippocampus of untreated age-matched mice [120]. Notably, the immunomodulatory effects of FAAH inhibition on the brain were described to be age-dependent. Indeed, in aged rats, the chronic pharmacological inhibition of hippocampal FAAH decreased microglial activation marker expression, pro-inflammatory cytokines and synaptic plasticity deficits compared to age-matched controls [121]. On the contrary, in the young animals, the same treatment had no effects [121]. All of these in vivo findings suggest that the impact on microglia exerted by FAAH inhibition may be strongly influenced by the type of stimulus from which the inflammation originates and/or the particular microenvironment surrounding microglial cells.

In vitro studies further support the idea that FAAH inhibition is effective in modulating microglia responses under different types of pro-inflammatory stimuli, even if they do not fully clarify the biochemical mechanisms underlying the observed effects. For instance, in rat primary microglia under LPS stimulation, URB597 reduced the expression of COX-2 and iNOS with a concomitant attenuation of the release of prostaglandin E2 (PGE2) and NO. The effect of URB597 on LPS-stimulated PGE2 release was not reversed by selective CB_1_ or CB_2_ receptor antagonists [122]. Likewise, in BV-2 cells, a murine microglial cell line, the inhibition of the activity of FAAH, along with its siRNA knockdown, suppressed the LPS-induced expression of ROS, PGE2, COX-2 and microsomal PGE synthase. Interestingly, the effects mentioned above were mediated neither by CB_1/2_ receptors nor PPARs [123]. In the same cellular line, after Aβ peptide pro-inflammatory stimulation, FAAH inhibition was shown to decrease the release of pro-inflammatory cytokines, switching to the resting phenotype and reducing cell migration by modulating the Rho signalling pathway [124]. Moreover, URB597 also promoted the phagocytic activity of BV-2, by inducing a robust reorganization of the cytoskeleton [124]. Unfortunately, the study did not investigate the receptors that were involved in this process, nor the underlying signalling cascades.

The different components of the microglial eCB signalling are summarized in Table 1, along with their main effects on microglia physiology.

### 4.2. Alteration of ECS in Alzheimer’s Disease

#### Human Studies

Specific alterations in eCB signalling were observed in AD patients. In particular, based on the postmortem Braak staging method [130], CB_1_ was upregulated in the earliest stages [131,132], and downregulated in the advanced stages of AD [102,132,133]. However, other studies found conflicting results regarding the CB_1_ receptor expression, which remained unaffected in AD patients [95,134,135]. In human brains, CB_2_ receptor expression was found to be positively correlated with Aβ_42_ concentration, amyloid plaque burden, levels of hyperphosphorylated tau and neuritic tangles, consistent with the hypothesis that activated microglia could contribute to the inflammatory process of AD [95,102,133,136]. Other reports showed that the increase in the level of CB_2_ receptor was more pronounced in severe AD when compared with age-matched controls or moderate AD subjects [136]. CB_2_ mRNA expression in peripheral blood mononuclear cells (PBMCs) showed no differences between AD cases and controls [98]. Interestingly, in the brains of AD subjects, high levels of CB_2_ were found to be nitrosylated, an effect of the increase in peroxynitrite radicals attributable to microglia activation [102].

A reduction in the levels of AEA and its precursor, NArPE, but not of 2-AG, was observed in the temporal cortex of AD patients [137,138]. Yet, no differences in AEA or 2-AG concentrations in the plasma of AD and healthy controls were detected in preliminary studies [98,99].

An early report also found an enhanced enzymatic activity in the hippocampus of AD human patients of the two metabolic enzymes for 2-AG, DAGL and MAGL [139]. In particular, increased DAGL expression (specifically the DAGLβ isoform) within hippocampal neurons and local microglia was positively correlated with pathological AD progression in postmortem studies [135]. Contrasting results were reported for FAAH. One group documented a reduction in FAAH activity within neuronal membrane fractions obtained from the frontal cortex of AD [137], while another study found no difference in FAAH protein expression within AD hippocampal samples compared to controls [135]. Notably, in PBMCs of AD subjects, an increase in FAAH mRNA expression was observed in patients with lower mini-mental state examination (MMSE) scores [140,141]. More recently, we documented that FAAH was overexpressed, also at the protein level, in circulating monocytes of AD patients, and their levels correlated with the severity of pathology [142].

The above-described AD-induced modulation of the ECS in the brain and periphery may occur at multiple levels: it could involve transcriptional and epigenetic mechanisms. In this context, our group has highlighted the importance of epigenetic mechanisms in the regulation of FAAH in PBMCs from subjects with late-onset AD. In particular, we found a reduction in DNA methylation at the FAAH promoter in AD subjects *versus* controls, which correlated with an increase in expression of FAAH both at mRNA and protein levels in those patients [141].

### 4.3. Preclinical Studies

AD animal models are transgenic mice overexpressing mutant variants of human APP that provoke the accumulation of Aβ peptides and AD-like symptomatology [143].To accelerate/worsen the onset and the course of the amyloidosis, other models were developed by co-overexpressing other AD-related proteins, such as presenilin 1, apolipoprotein E (ApoE) and TREM2. All of these different types of AD-like models developed microgliosis and cognitive impairment, but with different time points of onset [144]. In some of these models, the expression and distribution of ECS elements were found with profound modifications compared to healthy mice. 

In our work, we showed that in pre-symptomatic Tg2576, characterized by high levels of human mutant amyloid precursor protein APPSwe (Swedish mutation K670N/M671L), the localization and the signalling of CB_1_ was altered, despite the unchanged expression levels [145]. In APPSwe/PS1ΔE9, as a consequence of co-expressed human forms of APPSwe mutation and human PSEN1 lacking exon 9, numerous amyloid depositions were developed much earlier than aged-matched Tg2576 mice. In this model during the symptomatic phase, CB_1_ was found to have a significant reduction compared to wild-type mice [146].

The “non-psychotropic” cannabinoid receptor CB_2_ was found to be consistently upregulated in AD murine models, corroborating the results found in AD patients. In the brain of 5xFAD (co-expressing five common AD mutations: three associated with APP, Swedish, Florida, and London and two linked to PSEN1, the M146L and L286V) the increase in the expression of CB_2_ receptor occurred in specific brain areas characterized by intense inflammation and amyloid deposits [147]. Marked increases in CB_2_ levels have also been found in the microglia of the APP/PS1 model [148]. Additionally, there is evidence concerning the alteration of TRPV1: in the brain of 3xTg (APP Swedish, tau P301L, and PSEN1 M146V) the expression of the receptor was found downregulated [128].

Some reports documented alterations in the concentration of eCBs in different brain areas of AD-like models. For example, Kolfalvi et al. showed a marked reduction in AEA levels in the hippocampus, as well as in the prefrontal cortex, of Tg2576 mice [149]. Even when not significant, a reduction in AEA level in the hippocampus of aged mice of the same model was also found by our group [150]. Piro and colleagues found in the brain of APPSwe/PS1ΔE9 that levels of 2-AG were considerably increased [151]. Unfortunately, there is a lack of information about eCBs enzyme dysregulation since the evaluation of metabolic enzymes, in terms of expression and activity, has not been directly assessed.

### 4.4. Impact of Microglial Endocannabinoid Signalling in Alzheimer’s Disease 

Important information on the impact of the microglial ECS in AD has been obtained from preclinical studies performed on AD-like mice (Table 2). CB_1_ chronic activation by ACEA in APPSwe/PS1ΔE9 was effective in restoring cognitive dysfunction [152]. Unfortunately, this study neglected to address the impact of CB_1_ stimulation on microglia-driven neuroinflammation. Similarly, in a rat model where microglial activation and memory impairments were induced by Aβ injection, the activation of CB_1_ by WIN55,212-2 (a non-selective CB_1/2_ agonist) improved the cognitive deficit. Moreover, WIN55,212-2 prevented microglial activation in the cortex of Aβ-treated rats [102]. In the same model, another group showed that WIN55,212-2, acting through CB_1_ and CB_2_ receptors, significantly improved memory functions, decreasing the expression of some neuroinflammatory markers, such as TNF-α, activated caspase-3, and nuclear NFκB [153]. Additionally, the latter study did not address whether these effects could be ascribed to the cannabinoid-dependent modulation of microglial properties.

The direct involvement of microglia in the anti-AD effects of cannabinoid-based drugs was highlighted by several studies mainly focused on the CB_2_ receptor [125,148,154,155]. In the pre- and early symptomatic APPSwe/PS1ΔE9 mice, Aso and colleagues administered the selective CB_2_ agonist JWH-133 during both stages of the disease. The chronic stimulation of CB_2_ significantly improved learning and memory performances compared to vehicle-treated mice [148]. The long-term stimulation of CB_2_ also produced remarkable anti-inflammatory effects by decreasing microgliosis around the plaque and downregulating the expression of several pro-inflammatory cytokines, such as IL-1, IL-6, TNF-α and IFN-γ. Notably, these anti-inflammatory effects were associated with reduced oxidative stress damage and tau hyperphosphorylation in neuritic plaques [148]. Consistently, the chronic stimulation of CB_2_ in the early phase of AD-like symptomatology of Tg2576 markedly lowered COX-2 and TNF-α, and ameliorated memory deficit [126]. Interestingly, in the same model, the chronic treatment by WIN55,212-2 did not show the same effect listed above. Moreover, Tg2676 mice showed an increase in microglial cells compared with wild-type in the cortex. This increased density was not altered by chronic treatment with WIN55,212-2, while the selective agonist of CB_2_ JWH-133 decreased the number of reactive microglia [160].

Controversial evidence has emerged about the impact of the genetic ablation of the CB_2_ receptor on the symptomatology of amyloidosis in AD-like mice. On the one hand, in APPSwe/PS1ΔE9 mice, the deletion of CB_2_ reduced microglial activation and infiltration of macrophages. Furthermore, these mice expressed low levels of soluble Aβ_40/42_, pro-inflammatory chemokines, and cytokines, and displayed an improvement in cognitive impairment. Interestingly, in this study, the microglia around the plaque assumed a branched morphology, linked to a homeostatic or dystrophic state [125,154]. On the contrary, in J20 mice, an AD-like model expressing mutated APP (K670N/M671L/V717F), Koppel et al. found that CB_2_ lacking mice exhibited increased Aβ deposits in the cerebral cortex, hippocampus and enhanced plaque-associated microglia [160].

There a few studies regarding the possible role of TRPV1 in regulating microglial functions. In the hippocampus and cortex of the 3xTg mice, microglial cells showed distinctive phenotypic changes, such as more protrusions and longer branch length, suggesting a less-activated state. In this model, the activation of TRPV1 via capsaicin decreased amyloid and phosphorylated tau pathology, reversed memory deficit and promoted microglia activation, metabolism and autophagy [128]. Notably, impaired autophagic flux was correlated with ageing, characterized by immune senescence of macrophages [158]. In APP/PS1 mice, TRPV1 activation downregulated the release of pro-inflammatory IL-6, TNF-α, and increased autophagy, which promoted the clearance of Aβ [127].

Growing evidence has revealed that the eCBs catabolic enzymes, MAGL and FAAH, are promising targets for controlling microglia activities in the context of AD-related neuroinflammation. For example, in early-symptomatic mice of APPSwe/PS1ΔE9 and 5xFAD models, the pharmacological inhibition of MAGL with JZL184 led to the cognitive improvement and prevented neuroinflammation by reducing reactive microgliosis in the hippocampus [129,159]. Aparicio et al. demonstrated that FAAH^−/−^ mice exhibited an increase in the M1/M2 ratio and a decrease in microgliosis [156]. In particular, the absence of FAAH triggered an imbalance of the microglial phenotype towards an exacerbated pro-inflammatory state, as revealed by the increased M1 over M2 markers, although no significant differences were found in memory performance [156]. Finally, the genetic ablation of FAAH was found to reduce Aβ levels, neuritic plaques and gliosis in the 5xFAD model [158]. Furthermore, in the same model, FAAH^−/−^ mice showed that cortical DAM microglia specifically overexpressed phagocytic-related receptors, such as TREM2 and cathepsin D [157]. These data suggest a specific gene expression profile related to DAM molecular signature in microglia from 5xFAD/FAAH^−/−^ mice that provoked enhanced Aβ phagocytosis and clearance. The authors suggest that the enhancement of AEA signalling, by promoting an alternative DAM phenotype, could have a beneficial impact on amyloidosis [157].

Overall, these findings suggest that the ECS may have a regulatory role in microglial biology, both in normal and AD-related pathological conditions. Furthermore, substantial evidence has also been accumulated indicating that targeted manipulation of eCB signalling by cannabinoid-based medicines may represent a novel and valuable strategy for managing AD.

## 5. Conclusions

Microglia, the immunocompetent guardians of brain homeostasis, were recently shown to display an extremely heterogeneous phenotype across different regions of the brain as well as in distinct stages of AD, as they can exist in many different types and activation states, from neuroprotective to neurodestructive. A robust body of literature has consistently documented the functional relevance of the microglial ECS that could specifically drive the acquisition of an anti-AD phenotype by these cells, consisting of an enhancement in phagocytosis, chemotaxis, and the release of anti-inflammatory and/or pro-resolutory mediators. In particular, preclinical evidence demonstrates that the enhancement of eCBs signalling, obtained through an inhibition of the main eCB-degrading enzymes, exerts potent immunomodulatory effects on microglia-driven responses and might represent a promising therapeutic strategy for attenuating AD-related neurodegenerative processes and cognitive decline. 

Unfortunately, until now, knowledge of the direct modulation of microglia through ECS is mostly based on in vitro studies. Indeed, in the preclinical studies, several groups demonstrated that the modulation of the ECS in AD-like mice led to improved symptomatology. Since ECS-based drugs strongly act on the neuronal functional recovery, it is very difficult to establish to what extent these effects are attributable to a direct impact on microglial activity [81]. In this context, future and more MG-focused investigations performed using cell-selective inducible knockout models could help to define the relative contribution of microglial eCB signalling in the recovery of brain homeostasis under AD-related pathological conditions.

## Figures and Tables

**Figure 1 cells-11-01237-f001:**
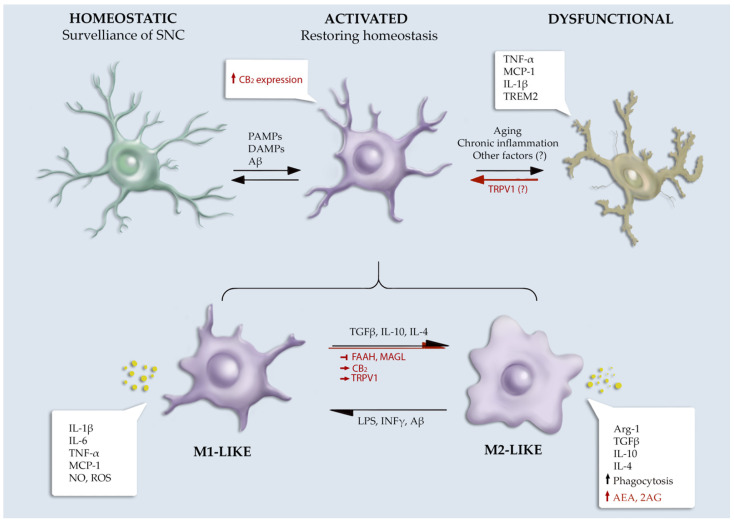
Schematic overview of ECS role in microglial phenotype. Symbols used: ↑, increased; ↓, decreased; →, activation; ⟞, inhibition.

**Table 1 cells-11-01237-t001:** ECS effects on microglia function in vitro.

ECS	Model	Pro-Anti-Inflammatory Challenge	Treatment	Effect on Microglia Function	Ref.
CB_1_	BV2	IFN-γ (100 U/mL)	SR141716A 1 µM	↑IFN-γ, IL-1β, IL-6, TNF-α, NO ↓MCP-1, CX3CL1	[93]
	Rat primary microglia	IL-4 IL-13 10 ng/mL each (M2a)	AM251 1 µM	↓Arg-1 immunostaining	[90]
CB_2_	Murine primary microglia	IFN-γ (100 U/mL) Aβ_42_ (1 µM)	JWH-015 5 µM	↑phagocytosis of Aβ_42_ ↓TNFα, NO	[103]
		LPS (100 ng/mL) IFNγ (20 ng/mL)	Constitutive KO	↓IL-6, TNFα =phagocytosis of Aβ_42_	[125]
		IL-4 (10 ng/mL) IL-13 (10 ng/mL)	Constitutive KO	↓phagocytosis	[90]
		Basal	2-AG (25 µM)	↑migration	[110]
	Rat primary microglia	Aβ_40_ soluble or fibrillar (500 nM)	HU-210, WIN55,212-2, JWH-133 (100 nM)	↓microglia activation (morphology) ↓TNFα	[102]
		LPS (100 ng/mL)	AEA (1 µM) AM-630 (0.1–0.5 µM)	↓NO ↓M1 phenotypic marker (mRNA TNFα, IL-β, IL-6, COX-2, iNOS)	[104]
		IL-4 IL-13 (10 ng/mL each)	AM630 1 µM	↓Arg-1 immunostaining	[90]
		LPS (10 ng/mL) IFNγ (10 U/mL)	AM1241 (10 µM)	↓IL-6, IL-β, iNOS ↑Arg-1, IL-10, BDNF, GDNF	[101]
	APP/PS1 glioma cells	Aβ40 (5 μg/mL)	WIN55,212-2, JWH-133 (200 nM)	↑Aβ transport through choroid plexus monolayers	[126]
	BV-2	LPS (50 ng/mL) IFNγ (100 U/mL)	AEA 5–15 µM (SR2 1 µM)	↑IL-10	[105]
TRPV1	Murine primary microglia	LPS (10 ng/mL) Aβ oligomer (5 µM)	Capsaicin (10 µM)	↑mTOR/AKT/HIF-1α pathway ↑phagocity of Aβ_42_	[127]
		LPS (100 ng/mL) ATP (5 mM)	Constitutive KO CPZ (10 µM)	↓NLRP3 inflammasome ↓IL-1β =TNF-α	[107]
		Basal	Capsaicin	↑TNF-α ↓IL-10	[88]
			Constitutive KO	↑IL-10	
			Capsaicin (10 µM)	↑migration	[87]
		Basal	Capsaicin (10 μM)	↑phagocytosis Aβ	[128]
		Basal	Constitutive KO	↑phagocytosis Aβ	
	BV2	basal	Capsaicin (10 μM)	↑phagocytosis Aβ	
	BV2	Phorbol myristate acetate (1 µM)	Capsazepine (50 µM)	↓ROS	[86]
MAGL	Microglia from adult brain	Aβ_42_ (10 μM) LPS (1 µg/mL) and IFN-γ (100 ng/mL)	JZL184 (1 μM)	↓NO, IL-1β (stimulation with LPS/IFN-γ) ↓Iba1 (stimulation with Aβ_42_)	[129]
FAAH	Rat primary microglia	LPS (0.03 µg/mL)	URB597 (10 µM)	↓COX-2, iNOS, PGE2	[122]
	BV2	Aβ _25–35_ (30 µM)	URB597 (5 µM)	↑cell viability ↓basal migration ↑phagocytosis ↑mRNA TGF-β, IL-10, ARG1 ↓mRNA TNF-α, IL-1 β, iNOS	[124]
		LPS (100 ng/mL)	URB597(10 µM) PF3845 (10 µM) siRNA	PF3845 ↓mRNA COX-2, IL-1 β, MCP1 PGE2, TNF-α URB597 ↓mRNA PGE2, IL-1 β, MCP1, siRNA ↓mRNA TNF-α, il-6, IL-1 β, MCP1 ↓COX-2, iNOS	[123]

Symbols used: ↑, increased; ↓, decreased; =, unchanged.

**Table 2 cells-11-01237-t002:** ECS immunomodulatory effect in AD mice models.

Model	ECS	Treatment	Molecular Effect	Behavioural Effect	Pre- Symptomatic	Early Symptomatic	Late Symptomatic	Ref.
APPSwe/PS1ΔE9	CB_2_	constitutive KO	=IL-6 ↓TNF-α and CCL2 ↓microgliosis, Iba1 in hipp ↓brain-infiltrating macrophage ↑ramified microglia around plaque ↓Aβ plaque in cx ↓Aβ plaque in hip	↑MWM			▼	[125,154]
		JWH-133 (0.2 mg/kg i.p.) 5 weeks	=Aβ burden in the cx =Aβ40 Aβ42 protein level	↑V-maze ↑Active avoidance test	▲ ▼			[148]
			↓TNF-α, IL-10, IL-6 IL-1β ↓microgliosis, Iba1 in cx (cells plaque associated) ↓tau phosphorylated ↓p38, GSK3β, SAPK/JNK ↓HNE =Aβ burden in the cx =Aβ40 Aβ42 protein level =Aβ plaque load	↑V-maze =Active avoidance test		▲ ▼		
		JWH-015 (0.5 mg/kg i.p.) 8 weeks	↓microgliosis, Iba1 in cx ↓mRNA TNF-α iNOS IL-6 = microgliosis, Iba1 in hipp =level of Plaque deposition	↑NOR =MWM		▲	▼	[155]
	TRPV1	capsaicin (standard chow plus 0.01% capsaicin) 4 weeks	=Aβ40, Aβ42 soluble fraction ↓Aβ40, Aβ42 insoluble fraction ↑autophagy ↑clearance of Aβ via autophagy (colocalization of iba1/LC3) ↓IL-6, TNF-α	↑MWM		▲ ▼		[127]
	MAGL	Constitutive KO	↓microglia, Iba1 ↓mRNA IL-1β, IL-6, TNF-α ↓Aβ plaques as well as the Aβ40, and Aβ42 amyloidogenic peptides			▼		[151]
		JZL184 (40 mg/kg i.p.) 2 weeks	↓mRNA IL-1β, IL-6, TNF-α			▲ ▼		
		JZL184 (40 mg/kg i.p.) 4 weeks	↓microgliosis, Iba1 in cx, hipp			▲ ▼		[129]
	CB_1_	ACEA (1.5 mg/kg i.p.) 4 weeks	=microglia activation, Iba1	↑V-maze	▲ ▼			[152]
5xFAD	FAAH	Constitutive KO	↑M1/M2 ratio in (FAAH^−/−^) ↓microgliosis Iba1 ↑mRNA IL-1β, TNF-α ↓mRNA IL-10, IL-4 ↓soluble Aβ42	=MWM		▼		[156]
			↑phagocytic Aβ by DAM			▼		[157]
		Constitutive KO	↑mRNA IL-1β, IL-6 =IL-6 ↑IL-1β in cx ↓microgliosis, Iba1 in hipp (cells plaque associated) ↓APP ↓Aβ42 and Aβ40	↑MWM		▼		[158]
		URB597 (3 mg/kg i.p.) 2 weeks	↑mRNA IL-6 in hipp =mRNA IL-1β and TNF-α in hipp	=MWM		▲ ▼		
	MAGL	JZL184 (12 mg/kg, i.p) 8 weeks	↓Aβ40 and Aβ42 as well as APP c-terminal fragments (CTFa/b) ↓reactive microglia, CD11b in hip	↑MWM		▲ ▼		[159]
J20	CB_2_	Constitutive KO	↑microgliosis, Iba1 in hipp (cells plaque associated) =microgliosis, Iba1 in hipp ↑soluble Aβ ↑Aβ plaque load =soluble Aβ40				▼	[160]
Tg2576		JWH-133 (drinking water at a dose of 0.2 mg/kg) 16 weeks	↓microgliosis, Iba1 in cx ↓COX-2 ↓CB2 protein ↓27% levels of Aβ40 ↓30% levels of Aβ42	↑NOR		▲	▼	[126]
	CB_1/2_	WIN 55,212-2 (drinking water at a dose of 0.2 mg/kg) 16 weeks	=microgliosis, Iba1 in cx =COX-2 ↓CB2 protein ↓30% levels of Aβ42	=NOR		▲	▼	
3xTg	TRPV1	Capsaicin (1 mg/kg i.p.) 4 weeks	↑microgliosis, Iba1 ↑autophagy ↑activated microglia in hipp and cx	↑Y Maze ↑MWM			▲ ▼	[128]
Rat (Aβ25-35 inj)	CB_1/2_	WIN 55,212-2 (10 µg intracerebroventricular injection) 1 week	↓microglia activation in cx	↑MWM				[102]
Rat (Aβ42 inj)			↓TNF-α ↓NF-kB	↑MWM				[153]

Symbols used: ▼, end of treatment/evaluation point; ▲, start of treatment. ↑, increased; ↓, decreased; =, unchanged; Abbreviations: MWM, Morris water maze; NOR, novel object recognition; hipp, hippocampus; cx, cortex.

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
