# Peer review of "Microglial Endocannabinoid Signalling in AD"

_cells, 2022, doi:10.3390/cells11071237_

Round 1

Reviewer 1 Report

This is an exceptionally well written manuscript, which will be extremely useful for scientists in Alzheimer research. I have only some comments and remarks

Remarks

Line 107: There is indeed an ongoing debate about the existence and importance of M1 and M2 states. However, the authors may add that these are the extreme endpoints of a spectrum of microglial physiological states, which exists in this pure form only in vitro in cell cultures but probably not in vivo.

Line 224: it is important to add that activated microglia contribute to tau pathology and beta-amyloid aggregation, key elements of AD. See:

Venegas C et al. Microglia-derived ASC specks cross-seed amyloid-β in Alzheimer’s disease. Nature 552, 355–361 (2017) and

Ising C et al: NLRP3 inflammasome activation drives tau pathology. ,Nature. 2019 Nov;575(7784):669-673.

Line 329: It is important to add that ECS influences microglial activity also indirectly, through neuronal CB1 receptors because microglial activity is under a strong neuronal (mostly inhibitory control). See: Ativie et al, Front.Mol.Neurosci. 2018, 11:295

The same issue appears at chapter 4.4 where the discussed pharmacological treatments, genetic modifications were not microglia specific. Thus, the observed changes in microglia phenotype could be the indirect consequence of an altered neuronal activity in the treated / genetically modified animals.

I appreciated the tables, which give a clear overview of the previous studies. To note: there is a typo in Table 2 at ref. 160.

Author Response

Point 1: Line 107: There is indeed an ongoing debate about the existence and importance of M1 and M2 states. However, the authors may add that these are the extreme endpoints of a spectrum of microglial physiological states, which exists in this pure form only in vitro in cell cultures but probably not in vivo.

Response 1: We thank the Reviewer for giving us the opportunity to implement this point. We briefly mentioned this issue in the text (lines 125-127).

Point 2: Line 224: it is important to add that activated microglia contribute to tau pathology and beta-amyloid aggregation, key elements of AD. See:

 Venegas C et al. Microglia-derived ASC specks cross-seed amyloid-β in Alzheimer’s disease. Nature 552, 355–361 (2017) and

Ising C et al: NLRP3 inflammasome activation drives tau pathology. ,Nature. 2019 Nov;575(7784):669-673.

Response 2: We thank the reviewer for this suggestion. We added further information in the text (lines 226-228).

Point 3: Line 329: It is important to add that ECS influences microglial activity also indirectly, through neuronal CB1 receptors because microglial activity is under a strong neuronal (mostly inhibitory control). See: Ativie et al, Front.Mol.Neurosci. 2018, 11:295

Response 3: We agree with the reviewer and briefly discuss this point in the text (lines 317-319).

Point 4: The same issue appears at chapter 4.4 where the discussed pharmacological treatments, genetic modifications were not microglia specific. Thus, the observed changes in microglia phenotype could be the indirect consequence of an altered neuronal activity in the treated / genetically modified animals.

Response 4: We also agree with these comments. We briefly discussed the limitations in the conclusions (lines 624-632).

Point 5: I appreciated the tables, which give a clear overview of the previous studies. To note: there is a typo in Table 2 at ref. 160.

Response 5: Thank you for your kind appreciation. Done, as suggested.

Reviewer 2 Report

Dear Editor,

The review paper submitted by Dr. Scipioni entitled “The Microglial Endocannabinoid Signalling in AD” and collaborators provides a comprehensive analysis of the most recent discoveries in endocannabinoids’ research, putting particular emphasis on the importance of the endocannabinoid system as a master regulator of several biological functions exerted by microglia during Alzheimer’s disease and/or in animal models. The review also offers important perspective on the use of putative drug candidates targeting the ECs to improve the inflammatory outcome in people afflicted by this debilitating disease.

The quality of the work is reflected by the thorough literature research performed by the authors to highlight the most recent work on ECs and microglia. However, there are some points that would require the authors’ attention to improve this work further. These points are listed as either MAJOR or MINOR below.

MAJOR

Figure 1 – This schematic picture is nicely represented. However, it is difficult to interpret the message. Does the picture portray the effects of CB2 stimulation on microglial phenotypic fate? If so, perhaps the authors might want to include a more detailed explanation of the illustration for an easier interpretation.

Has any of the ECS-targeting compounds ever reached clinical trials? Most of the work is focussed on preclinical models (which is understandable). However, it is also important to document any clinical advancement (or not) obtained using this class of drugs. It may be worth introducing a section showing where we are at, what drugs have been tested, outcomes and downfalls.

MINOR

English language – There are several typos throughout the text. I would encourage a detailed revision of the manuscript.

Table 2 (Header) – Please correct as follows: Pre-symptomatic, Early symptomatic, Late symptomatic

Table 2 (content) – Several typos (i.e. ramificated [RAMIFIED] microglia, n cells plaque asoociated [ASSOCIATED}, ↑ clarence [CLEARANCE] of Aβ, plaque [capitalised], cx or cortex [use either consistently], ↓microlgiosis [MICROGLIOSIS],Iba1in [NO SPACE] hipp, ↑activeted [ACTIVATED] microglia, ↓microglia activation in cortex [CORTEX])

Table 2 (caption) - ↑, increased; ↓, decreased; =, same [UNCHANGED];

Author Response

Point 1: Figure 1 – This schematic picture is nicely represented. However, it is difficult to interpret the message. Does the picture portray the effects of CB2 stimulation on microglial phenotypic fate? If so, perhaps the authors might want to include a more detailed explanation of the illustration for an easier interpretation.

Response 1: We thank the reviewer for this suggestion. We add a more detailed caption of the picture.

Point 2: Has any of the ECS-targeting compounds ever reached clinical trials? Most of the work is focussed on preclinical models (which is understandable). However, it is also important to document any clinical advancement (or not) obtained using this class of drugs. It may be worth introducing a section showing where we are at, what drugs have been tested, outcomes and downfalls.

Response 2: We thank the reviewer for this suggestion. To date, through the modulation of ECS only few clinical trials have been performed and reported several beneficial effects in AD-related symptoms. Regardless of their importance, none of them could evaluate a possible effect on microglia. This is probably due to that neuroinflammation is demonstrated to have a key role on the progression to the onset of Alzheimer’s disease only in recent years.

Point 3: English language – There are several typos throughout the text. I would encourage a detailed revision of the manuscript.

Response 3: We apologize for the many inaccuracies and mistakes that we have carefully amended.

Point 4: Table 2 (Header) – Please correct as follows: Pre-symptomatic, Early symptomatic, Late symptomatic

Response 4: Done, as suggested.

Point 5: Table 2 (content) – Several typos (i.e. ramificated [RAMIFIED] microglia, n cells plaque asoociated [ASSOCIATED}, ↑ clarence [CLEARANCE] of Aβ, plaque [capitalised], cx or cortex [use either consistently], ↓microlgiosis [MICROGLIOSIS],Iba1in [NO SPACE] hipp, ↑activeted [ACTIVATED] microglia, ↓microglia activation in cortex [CORTEX])

Response 5: Done, as suggested.

Point 6: Table 2 (caption) - ↑, increased; ↓, decreased; =, same [UNCHANGED];

Response 6: Done, as suggested.

Round 2

Reviewer 2 Report

Dear Editor,

The authors have partly addressed the issues raised by this reviewer and the paper has improved in quality.

However, some of the typos identified in the text/tables were not amended correctly:

  1. TABLE 2: It should be "Early symptomatic" and "Late symptomatic" NOT syntomatic. "Clearence" should be corrected and replaced with the term "clearance".
  2. Grammar/language corrections were not highlighted in the manuscript